# Endoscopic Endonasal Resection of the Medial Wall of the Cavernous Sinus and Its Impact on Outcomes of Pituitary Surgery: A Systematic Review and Meta-Analysis

**DOI:** 10.3390/brainsci12101354

**Published:** 2022-10-06

**Authors:** Leonardo J. M. de Macêdo Filho, Ana Vitória G. Diógenes, Esther G. Barreto, Bhavya Pahwa, Susan L. Samson, Kaisorn Chaichana, Alfredo Quinones-Hinojosa, Joao Paulo Almeida

**Affiliations:** 1Health Science Center, University of Fortaleza, Av. Washington Soares 1321, Fortaleza 60811-905, Ceará, Brazil; 2Department of Neurosurgery, Mayo Clinic, 4500 San Pablo Rd, Jacksonville, FL 32224, USA; 3Department of Medicine, University College of Medical Sciences, 2, Tahirpur Rd, GTB Enclave, Dilshad Garden, New Delhi, Delhi 110095, India; 4Department of Medicine, Mayo Clinic, Jacksonville, FL 32224, USA

**Keywords:** medial wall resection, cavernous sinus, pituitary adenomas, neurosurgery, endoscopic

## Abstract

Introduction. Pituitary adenomas have the potential to infiltrate the dura mater, skull, and the venous sinuses. Tumor extension into the cavernous sinus is often observed in pituitary adenomas and techniques and results of surgery in this region are vastly discussed in the literature. Infiltration of parasellar dura and its impact for pituitary surgery outcomes is significantly less studied but recent studies have suggested a role of endoscopic resection of the medial wall of the cavernous sinus, in selected cases. In this study, we discuss the techniques and outcomes of recently proposed techniques for selective resection of the medial wall of the cavernous sinus in endoscopic pituitary surgery. Methods. We performed a systematic review of the literature using the Preferred Reporting Items for Systematic Reviews and Meta-Analyses (PRISMA) guidelines and protocol and a total of 4 studies with 106 patients that underwent an endoscopic approach for resection of pituitary tumors with resection of medial wall from cavernous sinus were included. Clinical and radiological data were extracted (sex, mean age, Knosp, prior surgery, tumor size and type, complication rate, and remission) and a meta-analysis using the RevMan 5.4 software was performed. Results. A total of 5 studies with 208 patients were included in this analysis. The mean age of the study population was 48.87 years (range 25–82) with a female/male ratio of 1:1.36. Majority of the patients had Knosp Grade 1 (*n = 77*, 37.02%) and Grade 2 (*n = 53*, 25.48%). The complication rate was 4.81% (*n = 33/106*) and the most common complication observed was a new transient CN dysfunction and diplopia. Early disease remission was observed in 94.69% of the patients (*n = 196/207*). The prevalence rate of CS medial wall invasion varied from 10.4 % up to 36.7%. This invasion rate increased in frequency with higher Knosp Grade. The forest plot of persistent disease vs. remission in this surgery approach showed a *p* < 0.00001 and heterogeneity (I^2 = 0%). Discussion. Techniques to achieve resection of the medial wall of the cavernous sinus via the endoscopic endonasal approach include the “anterior to posterior” technique (opening of the anterior wall of the cavernous sinus) and the “medial to lateral” technique (opening of the inferior intercavernous sinus and). Although potentially related with improved endocrinological outcomes, these are advanced surgical techniques and require extensive anatomical knowledge and extensive surgical experience. Furthermore, to avoid procedure complications, extensive study of the patient’s configuration of cavernous ICA, Doppler-guided intraoperative imaging, surgical navigation system, and blunt tip knives to dissect the ICA’s plane are recommended. Conclusion. Endoscopic resection of the medial wall of the cavernous sinus has been associated with reports of high rates of postoperative hormonal control in functioning pituitary adenomas. However, it represents a more complex approach and requires advanced experience in endoscopic skull base surgery. Additional studies addressing case selection and studies evaluating long term results of this technique are still necessary.

## 1. Introduction

Pituitary adenomas, although histologically benign and more commonly restricted to the sella space, have the potential to invade parasellar structures, infiltrating the dura mater, cranium, and the venous sinuses [1].

Surgery is the main treatment for most pituitary tumors, leading to disease control in 60–90% of cases [1,2,3,4]. Many factors contribute to the heterogeneity of results in pituitary surgery, including tumor volume, tumor subtype, previous surgery, and tumor extension into the cavernous sinus. The involvement of surrounding structures is associated with higher frequency of incomplete resection and recurrence, especially for macroadenomas [5]. Additionally, to lower chances of disease control, larger tumors and those with invasion of the cavernous sinus are more prone to complications, including diabetes insipidus and hypopituitarism, cerebrospinal fluid leak, meningitis, loss of vision, carotid artery injury, and death [1]. Therefore, tumors with extensions to the parasellar space often are not completely resected and are treated with multimodal management, with a combination of subtotal resection followed by clinical follow up and/or medical and/or radiation therapy, to maximize chances of tumor control while reducing chances of complications [6].

Understanding of the anatomy of the cavernous sinus (CS) and parasellar region has led to the development of new surgical techniques and improvement in surgical results [7]. The walls of the CS can be divided into four parts: superior/roof, posterior, medial, and lateral. The medial wall is composed by the dura mater’s inner layer/meningeal layer that extends from the diaphragm and front of the sella. It shelters the pituitary gland and separates it from the ICA cavernous segment and the venous drainage, and forms the lateral boundary of the hypophyseal fossa. Understanding of the anatomy and display of anchoring patterns, such as the parasellar ligaments, is fundamental to perform its removal more safely [8]. 

Due to its proximity to the pituitary gland and sella, the medial wall of the CS is one of the structures most commonly invaded by these tumors. Such tumor extension has been considered as a major factor associated with persistent disease and recurrence for pituitary adenomas located laterally in the sella or extending to the cavernous sinus. Therefore, surgical resection of the medial wall of the cavernous sinus has been proposed [7,9,10,11] for resection of selected pituitary adenomas, with improved remission rates as well as minimal complications and recurrence rates [12]. This technique has been popularized and further applied in more recent years, however there is a paucity of clinical studies in the literature to validate its results. 

In the current study, we review the current clinical results of endoscopic pituitary surgery with use of resection of the medial wall of the cavernous sinus. Surgical techniques, indication, results, and complications are reviewed and discussed to provide an updated overview of this technique and its applications in modern pituitary and skull base surgery. 

## 2. Methods

### 2.1. Literature Search 

We performed a systematic review of the literature using the Preferred Reporting Items for Systematic Reviews and Meta-Analyses (PRISMA) guidelines and protocol (Figure 1). A literature search was performed using PubMed, EMBASE, Ovid, and SCOPUS databases up to 30 August 2022. Search terms included (medial wall) AND (cavernous sinus) AND ((removal) OR (resection)). We selected full-text articles published from January 1990 to August 2022. Screening of titles and abstracts was performed, and further evaluation of full-text publications was used to further exclude studies. 

### 2.2. Study Selection

Inclusion criteria were case series in which patients underwent an endoscopic approach to pituitary adenomas with resection of medial wall of the cavernous sinus due to tumor invasion. Exclusion criteria were patients with preservation of cavernous sinus medial wall. Included studies were assessed by two authors (L. J. M. M. F and J. P. C. A) to ensure that cases were correctly included in the study. Patient data from multiple studies were combined into one table for comparison (Table 1).

### 2.3. Data Extraction and Statistical Analysis 

We extracted details on the patient characteristics including age, sex, previous surgery, and type of tumor, Knosp Grade, tumor size, medial wall invasion, and endocrinologic remission. We performed a meta-analysis using the RevMan 5.4 software (RevMan 5.4, Review Manager, Version 5.4, The Cochrane Collaboration, 2020). The mean difference was used as a measure of effects, with confidence intervals of 95%, and the results of the recurrence and remission groups were compared by the statistical method of inverse variance. Statistical heterogeneity was assessed using Chi2 statistics and I2 scores in a forest-plot (Figure 2). 

## 3. Results

### 3.1. Study Selection

We initially retrieved 582 studies after duplicate removal: 77 records were retrieved from PubMed, 120 from Embase, 392 studies from Ovid, and 341 from Scopus, totaling 349 full texts screened. Ten full-text articles were assessed for eligibility; five studies were excluded due to preservation of the medial wall and 1 study due to incomplete demographic and tumor characteristics data. Therefore, 5 remaining studies were selected for data extraction (Figure 1).

A total of 5 studies with 208 patients were included in this analysis. The mean age of the study population was 48.87 years (range 25-82) with a female/male ratio of 1:1.36. The majority of the patients had Knosp grade 1 (*n = 7*, 37.02%) and grade 2 (*n = 53*, 25.48%) followed by grades 0 (*n = 42*, 20.19%), 3 (*n = 31*, 14.9%), and 4 (*n = 5*, 2.4%).

We compiled patient and tumor characteristics of all studies (sex, mean age, Knosp, prior surgery, tumor size and type, complication rate, and remission) in Table 1. The tumor size classification was not reported in all studies. One patient was lost to follow-up, thus 207 of 208 subjects had remission evaluated properly after surgery. The forest plot of persistence of disease vs. remission in this surgery approach showed a *p* < 0.00001 and heterogeneity (I^2^ = 0%) (Figure 2).

### 3.2. Demographics and Tumor Characteristics of Included Studies

Omar [8] reported sixteen patients that underwent resection of the medial cavernous wall in addition to tumor excision; thirteen patients were females, and the mean age was 40.9 ± 15.4 years. The largest diameter (mean) was 1.5 ± 0.68 cm, moreover, eleven patients had macroadenomas and five microadenomas. The tumor types were five ACTH adenomas, five prolactinomas and six GH adenomas. Radiologically, three cases were KNOSP grade 0, two were grade 1, six were grade 2, two were grade 3, and three were grade 4. Only two patients (13%) had history of previous surgery. The authors resected the medial wall of the cavernous using a technique that consisted of opening of the inferior sellar dura and then separation of the anterior and medial walls of the cavernous sinus following a medial to lateral dissection. The mean blood loss was 184 (100–400) mL, with none of the patients requiring blood transfusion. Most of the subjects (13 patients - 81%) underwent gross total resection and three patients subtotal resection. Fifteen patients had no recurrence or stable residual tumor (disease control) and one patient had progression/recurrence. The complication rate in this study was 0%. The median length of follow-up was 11 (range 1–70) months [8].

Nagata [12] performed medial wall removal, using a similar technique as Omar et al, in 14 patients, seven females and 7 males, with a mean age of 57.9 (range 29–82) years. The mean largest diameter was 1.3 (0.51–1.82) cm. Three tumors were consistent with microadenoma and 11 were macroadenomas. Tumor types were 12 GH adenomas and 2 ACTH adenomas. On the Knosp classification scale, two tumors were grade 0, six were grade 1, four were grade 2, and two were grade 3. Only one patient (7.14%) had had a previous surgery. The mean intraoperative blood loss was 170 (range 32–400) mL, and any blood transfusion. Early biochemical remission was achieved in 13 patients (92.9%); only one patient failed to achieve early remission. Thirteen patients had remission during the follow-up period and one patient (7.14%) showed recurrence. One patient showed transient cranial nerve (CN) III palsy as a complication of the procedure. The median length of follow-up was 11.8 (range 4–21) months [12].

Cohen-Cohen [13] reported a case series of resection of the medial wall of the cavernous sinus in pituitary surgery. Fifty cases of medial wall resection were performed, using a technique where dissection was directed from anterior to posterior, with opening of the anterior wall of the cavernous sinus as the initial step, then followed by section of ligaments between the medial wall and cavernous carotid and then by resection of the medial wall. The female/male ratio was 1:1, with a mean age of 49 years. The mean largest tumor diameter was 1.59 (range 0.5–3.8, median 1.5) cm. There were 17 microadenomas and 33 macroadenomas. The tumor types were 15 nonfunctional, 16 GH adenomas, 10 prolactinomas, and 9 ACTH adenomas. They reported that 11 cases were KNOSP grade 1, 23 were grade 2, and 16 cases were grade 3. Five patients (10%) had prior history of adenoma resection. The average blood loss was 378 (range 50–1200) ml, and two patients required blood transfusion. Early biochemical remission was achieved in all patients of this study. Only one patient showed recurrence after 2 years; at the last follow-up, the remission rate was 97% (34 of 35 cases). Five patients had complications, one patient had CSF leak, and four developed a new CN palsy, three of them partial CN IV palsy and one CN III, all of them were resolved after 3 months follow-up. The mean follow-up was 30 (range 4–64) months for patients with functional adenomas and 16 (range 4–30) months for those with nonfunctional adenomas [13]. 

Mohyeldin [14] evaluated the effects of medial wall resection, using the technique reported by Cohen-Cohen et al, on biochemical remission in acromegaly. 26 patients were included in this study, 11 were female and 15 were male, the mean age was 49.3 ± 15.5 years. Twenty-one patients (81%) underwent the medial wall resection. The average tumor size (maximum diameter) was 1.842 ± 0.982 cm. WHO adenoma classification (2017) presented 4 patients with densely granulated, 10 sparsely granulated, 7 mixed somatic lactotroph, 2 mammosomatotroph, and 3 plurihormonal tumors. Six tumors are classified as KNOSP grade 0, twelve were grade 1, four were grade 2, two were grade 3B, and two were KNOSP grade 4. All patients are newly diagnosed acromegaly, and none had prior surgery. Early remission was achieved in 23 of 25 patients (92%); two patients after 6 months of follow-up presented biochemical recurrence and needed adjuvant treatment, one patient was lost to follow-up. Four patients experienced postoperative transient diplopia (CN VI), two of whom recovered within one week, while the other two recovered by the one-month follow-up; moreover, two had a postoperative CSF leak requiring surgical repair. The mean follow-up for the acromegaly cohort was 15.56 (range 3–30) months [14]. 

Ishida [15] presented the largest case series of functioning pituitary tumors; the invasion of MW of CS was diagnosed by an endocrine pathologist, and patients with possible wall invasion and clear CS invasion underwent medial wall resection surgery. 248 patients were included in this study but only 102 (41.13%) patients had the MW resected due CS invasion; sixty-four (62.7%) were female and thirty-eight (37.3%) were male. The mean age of these groups was 48.7 (±14.1) years. Fifty (49%) tumors were consistent with microadenoma and 52 (51%) were macroadenomas. Tumor types were 67 (65.7%) GH adenomas, 10 (9.8%) prolactinomas, 2 (2%) TSH-secreting adenomas, and 23 (22.5%) ACTH adenomas. On the Knosp classification scale, 31 tumors were grade 0, 46 were grade 1, 16 were grade 2, and 9 were grade 3. Eleven patients (10.8%) had had a previous surgery. Early remission was achieved in 96 of 102 patients (94.12%). There were not any complications observed in the studied group. The patients had been followed up in 1 year [15]. 

## 4. Discussion 

Complete resection and endocrinological remission of pituitary adenomas depends on the tumor size, histological subtype, and presence of invasion of surrounding structures. Pituitary adenomas with invasion of parasellar tissue, especially into the CS, have an increased risk for incomplete resection and recurrence [13,16]. The prevalence rate of CS invasion varies from 10.4 % up to 36.7% [13,16]. This invasion rate increased in frequency with higher Knosp grade [16,17]. Commonly, there is involvement of CS medial wall with preservation of CS compartments [9] and it is often a unilateral invasion [13]; its detection, however, is only possible through surgical direct observation or signs in previous imaging due to the close relationship with the internal carotid artery (ICA), therefore current imaging studies are not sufficient for preoperative evaluation of invasion of the medial wall of the cavernous sinus [17]. The variability in the shape, size, and distribution of the venous plexus makes the identification harder on magnetic resonance, which makes intraoperative confirmation necessary [16]. Some techniques could help to confront these challenges by helping differentiate the target tumor from normal surrounding tissue, such as neuronavigation and intraoperative MRI; however, both methods have limitations. Therefore, there is the need to develop other techniques to help augment tumor resection, such as the use of fluorescence-guided surgery, that has proved its feasibility and safety, but still needs proofing of its applicability in real practice, cost-effectiveness assessment, and impact in postoperative outcomes [18,19]. 

The natural course of cavernous sinus invasion by pituitary tumors was first explained in 1948 and shows that it initiates by the medial wall and, subsequently, invades other sites of the CS dural enclosure, intracavernous portion of the ICA adventitia, and finally to the surface of the cranial nerves that passes through the CS [9]. Therefore, since the medial wall is the primary site of invasion, its resection, in selected cases, is important for clinical outcomes. However, its resection is also challenging due to the anatomic proximity of the medial wall to critical neurovascular structures, including the internal carotid artery and venous plexi. Hence, criticism of the approach includes its potential higher chance of significant blood loss, carotid injury, and cranial nerve dysfunction [16]. Current data, however, suggests that this is a safe approach in selected cases, when performed in center with experience and correct instrumentation and anatomical knowledge, not only in pituitary but also cavernous sinus surgery. Different maneuvers can be performed to maximize control of the cavernous sinus and decrease venous blood loss, including raising the patient’s upper body by 30 degrees before performing the resection of the medial wall [12]. Additionally, hemostatic agents can be utilized to provide hemostasis at the time of deep attachment of the medial wall. To avoid injury to the ICA, a micro-Doppler is recommended, so the precise location of the cavernous ICA can be identified; additionally, blunt dissectors are recommended while use of scissors for excision of the parasellar ligaments is minimized. Adequate venous control facilitates the visualization of the anatomical landmarks: carotid clinoid ligament, the inter clinoid ligament, the posterior clinoid, the posterior gland, and the sellar diaphragm, allowing dissection of medial wall beyond the tumor invasion area [13]. 

Resection of the medial wall of the cavernous sinus via endonasal endoscopic approach has been documented as a safe and technically feasible method when performed by highly experienced neurosurgeons. In our analysis, we observed that the resection of CS medial wall was associated with low complications rates. Mohyeldin et al. (2022) reported it in 50 out of 107 patients with consecutive primary pituitary adenomas, in which four patients experienced postoperative transient diplopia and two patients experienced a postoperative CSF leak [14]. Nagata et al. (2019) removed the medial wall of one side of 14 patients and only one patient developed delayed right oculomotor nerve palsy and had the symptom completely resolved after oral steroid therapy [12]. Cohen-Cohen et al. reported removal of medial walls in 50 patients and four patients developed transient CN palsy; nonetheless, all patients obtained reversibility of the symptoms, which shows minimal morbidity configured by this additional surgical step [13]. They reported postoperative external ophthalmoplegia attributed to excessive packing of CS. 

Currently available studies report disease remission rates up to 97%. Those results are better than overall results reported in pituitary surgery; however, the median follow-up varied from 11 to 30 months, which is a relatively short time for follow-up in pituitary adenomas and does not allow for long term extrapolations. Sample size in these studies were also relatively small, which can be explained by the recent dissemination of this surgical technique and the need for experienced team surgery. 

Most authors conclude that having extensive surgical experience and vast neuroanatomy knowledge is essential and there is overall agreement that it is a useful technique in selected cases where tumor invasion of the medial wall is observed. The application of this technique in functioning adenomas (GH, ACTH, TSH, prolactinomas) is more well-received, considering the morbidity of those tumors. Its application in non-functional tumors is more debatable as those tend to be slow-growing tumors and potential residual tumor in the medial wall of the cavernous sinus can often be followed clinically with no additional morbidity and treated with additional surgical debulking or radiosurgery if growth is observed. Maximum tumor resection and minimization of surgical complications are crucial goals. When discussing whether to elect and perform resection of the medial wall of the cavernous sinus, surgical teams should take into consideration their intimacy with cavernous sinus surgery and anatomy and perform careful assessment of the internal carotid anatomy. Correct instrumentation is mandatory and includes use of Doppler-guided intraoperative imaging, surgical navigation system, and blunt-tip knives to dissect the ICA’s plane [9,13,15,16,20]. This novel technique has the potential to become the gold standard treatment in the management of selected pituitary adenomas. 

## 5. Conclusions 

The resection of the medial wall of the cavernous sinus in selected cases of pituitary adenomas has been linked to high rates of disease control and positive outcomes in short-term follow-up. Despite previous concerns about higher complication rates, this procedure has shown low morbidity rates. This can be explained by selected use of this technique by teams with experience with cavernous sinus surgery and anatomy, adequate case selection based on preoperative radiological studies, and proper use of intraoperative tools, such as Doppler-guided imaging and surgical navigation system. Medial wall removal is especially useful in cases of functioning adenomas where the tumor extends to the medial wall of the CS and in cases of recurrent non-functional adenomas that present with close relationship to the cavernous ICA or clearly infiltrate the medial wall. 

## Figures and Tables

**Figure 1 brainsci-12-01354-f001:**
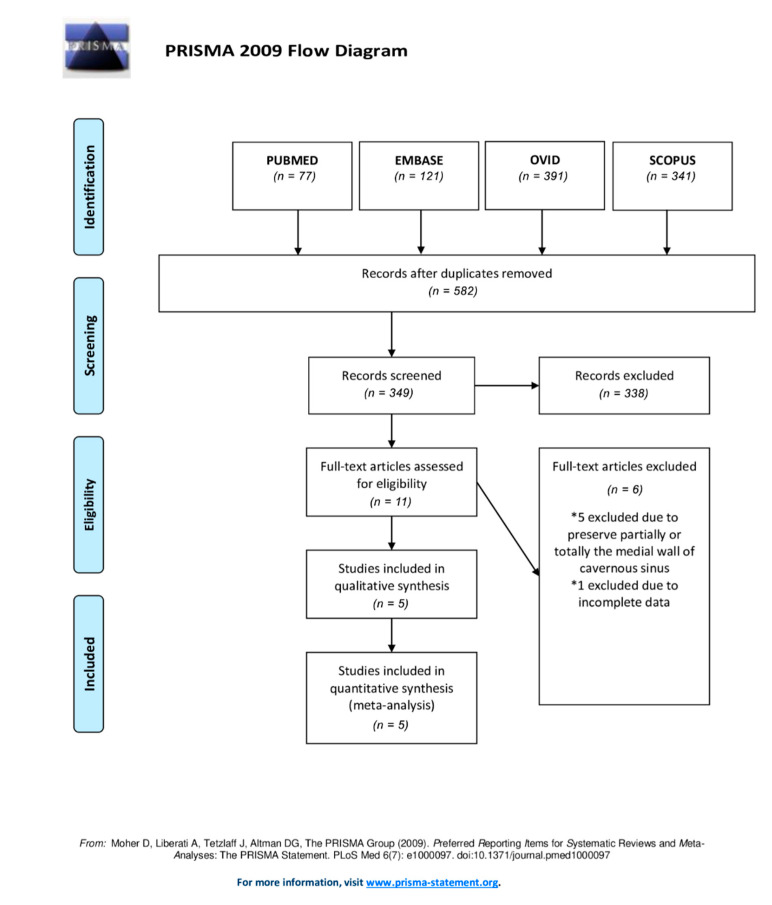
Flow diagram of Preferred Reporting Items for Systematic Reviews and Meta-Analyses (PRISMA) guidelines and protocol.

**Figure 2 brainsci-12-01354-f002:**
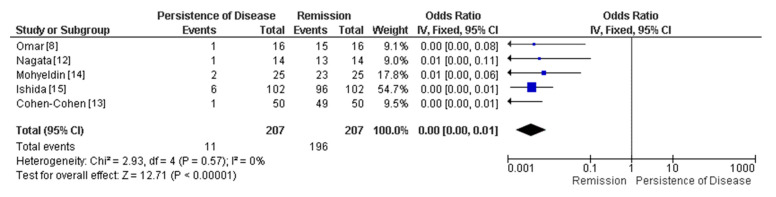
Forest plot of remission vs. persistence of disease of the 4 selected studies included in this analysis.

**Table 1 brainsci-12-01354-t001:** Patient cohort summary of the 5 selected studies included in this analysis. Variables include sex, mean age, Knosp grade, previous surgery, tumor size, tumor type, complication rate, and remission. Categorical variables are reported as number and percentage. (*) Not all studies described the tumor size. (§) Early remission; one patient didn’t continue the follow-up.

Patient and Tumor Characteristics	
**Sex**	**Number of cases (%)**
Male	88 (42.31%)
Female	120 (57.69%)
Mean Age	48.87
**Knosp Grade**	
Grade 0	42 (20.19%)
Grade 1	77 (37.02%)
Grade 2	53 (25.48%)
Grade 3	31 (14.90%)
Grade 4	5 (2.40%)
Previous Surgery	19 (9.13%)
**Tumor Size ***	
Macroadenoma	108 (59.34%)
Microadenoma	74 (40.66%)
Tumor Type	
**Non-functional**	15 (7.21%)
ACTH adenoma	39 (18.75%)
Prolactinoma	25 (12.02%)
GH adenoma	127 (61.065%)
TSH-secreting adenoma	2 (0.96%)
Complication Rate	10/208 (4.81%)
Remission §	196/207 (94.69%)

## Data Availability

Not applicable.

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
