# Peer review of "Endoscopic Endonasal Resection of the Medial Wall of the Cavernous Sinus and Its Impact on Outcomes of Pituitary Surgery: A Systematic Review and Meta-Analysis"

_brainsci, 2022, doi:10.3390/brainsci12101354_

Round 1

Reviewer 1 Report

The present review about the resection of CS MW is well written and conducted; the methods sound and results are interesting although obviuosly awaited; Going trhough the manuscript I found few points that the authors could improve as follow:

In my opinion the authors should briefly discuss the role of perioperative endoscopic fluoro-angiography in improving the tumoraldetection and excision especially in very eloquent area (please add literature and and discuss).

Pertinent literature is missing:  Chibbaro S, Signorelli F, Milani D, Cebula H, Scibilia A, Bozzi MT, Messina R, Zaed I, Todeschi J, Ollivier I, Mallereau CH, Dannhoff G, Romano A, Cammarota F, Servadei F, Pop R, Baloglu S, Lasio GB, Luca F, Goichot B, Proust F, Ganau M. Primary Endoscopic Endonasal Management of Giant Pituitary Adenomas: Outcome and Pitfalls from a Large Prospective Multicenter Experience. Cancers (Basel). 2021 Jul 18;13(14):3603. doi: 10.3390/cancers13143603. PMID: 34298816; PMCID: PMC8304085. (please add and discuss).

Reviewer 2 Report

Authors present a systematic review on impact of resection of medial wall (MW) of the cavernous sinus (CS) on outcome in pituitary surgery. The title of the study is misleading, since the authors state in the abstract that they have included only 4 studies with 106 patients who underwent endoscopic resection, so the title should be changed accordingly - either that, or the review should be broadened with studies on transsphenoidal and transcranial surgery.  Furthermore, the authors often write pituitary surgery and pituitary adenoma- these two are not sinonmys, since pituitary surgery includes lesions other than adenoma. The prevalence rate  of CS MW invasion varied from 10.4% up to 36.7% with complication rate of 9.43%. 

Authors conclude that  endoscopic resection of the MW of the CS has been associated with reports of higher rates of postoperative hormonal control in  functioning pituitary adenomas, when compared to previous series and other techniques. I suggest to delete this sentence, since this cannot be a conclusion following review of only 4 endoscopic studies - especially since the authors performed only a meta-analysis of the four studies and did not perform any kind of comparison to the surgical studies of transsphenoidal microscopic and transcranial microscopic technique. Although endoscopic surgery has been broadly advocated in the past 20 years, recent reviews have shown that only 20-25% of skull base surgeons have transffered to endoscopy alone - I suggest to comment on this in the Discussion. One further difficulty with this study is - the role of the attentional leave of tumor remnant in cavernous sinus, in order not to risk the vascular damage and postoperative deficits. This has become almost a standard which has shown better functional clinical outcome for the patient - regardless in which technique the surgery is performed - so that the role of surgery in CS alone is currently less and less important as the stereotactic radiosurgery of tumor remnant and recently particle therapy evolve. 

The authors have excluded the studies with mixed cohorts, which is a shame and does not fulfill the criteria of the systematic review - there are several studies which include a large number of patients who underwent endoscopic surgery for pituitary lesions where CS involvement was thematized. 

Furthermore, I suggest to include recent study into the systematic review and discuss:

Ishida A, Shiramizu H, Yoshimoto H, Kato M, Inoshita N, Miki N, Ono M, Yamada S. Resection of the Cavernous Sinus Medial Wall Improves Remission Rate in Functioning Pituitary Tumors: Retrospective Analysis of 248 Consecutive Cases. Neurosurgery. 2022 Aug 24. doi: 10.1227/neu.0000000000002109. Epub ahead of print. PMID: 36001781.

I suggest - if the intention is to show superiority of the endoscopic approach to microscopic approach - to perform a review which involves microscopic series and combined series and to perform needed statistical analysis. If the intention is only to summarize the experience of the endoscopic surgery of pituitary adenomas, then you should include mixed studies. 

Round 2

Reviewer 2 Report

The authors  have made significant revision to the manuscript, included the suggested study into their analysis and changed the title of the study.